# Black Sea Turtle (*Chelonia mydas agassizii*) Life History in the Sanctuary of Colola Beach, Michoacan, Mexico

**DOI:** 10.3390/ani13030406

**Published:** 2023-01-25

**Authors:** Bedolla-Ochoa Cutzi, Reyes-López Miguel Angel, Rodríguez-González Hervey, Delgado-Trejo Carlos

**Affiliations:** 1Centro de Biotecnología Genómica, Instituto Politécnico Nacional, Reynosa C.P. 88710, Tamaulipas, Mexico; 2Instituto de Investigaciones sobre los Recursos Naturales, Universidad Michoacana de san Nicolás de Hidalgo, Morelia C.P. 58330, Michoacán, Mexico; 3Centro Interdisciplinario de Investigación para el Desarrollo Integral Regional, Unidad Sinaloa, Instituto Politécnico Nacional, Guasave C.P. 8149, Sinaloa, Mexico

**Keywords:** black turtle, *Chelonia mydas agassizii*, body size, age of sexual maturity, clutch size, eastern Pacific

## Abstract

**Simple Summary:**

This paper presents important aspects of the life history of the black sea turtle (*Chelonia mydas agassizii*) population nesting on the beaches of Michoacan, Mexico. Information on morphometric and reproductive traits related to the life history of black sea turtles (body size, clutch size, egg size, fecundity, remigration interval, age at sexual maturity, and growth rate) was studied. An analysis of interannual variations in the life history traits of the species is also carried out, and the results obtained indicate that *C. m. agassizii* differs from the information reported for other populations of *Chelonia mydas mydas* distributed pantropically.

**Abstract:**

Sea turtles present strategies that have allowed them to survive and reproduce. They spend most of their lives in the sea, except when they emerge as hatchlings from the nest and when the adult females return to nest. Those moments of their life cycle are vital for their reproductive success, conservation, and knowledge of their biology. This study reports the life history traits exhibited by female black sea turtles from Colola Beach, Mexico using morphometric and reproductive data obtained during 15 sampling seasons (1985–2000, n = 1500). The results indicate that nesting females have a mean body size of 85.7 cm and reach sexual maturity at 24 years old at a minimum size of 68 cm. Females deposit a mean of 69.3 eggs per clutch, and the mean fecundity was 196.4 eggs per female per season. The remigration intervals of 3 and 5 years were the most frequent registered. The life history traits found in the black sea turtle population present the lowest values reported with respect to studies conducted in the Atlantic and Indo-Pacific green turtle populations, which supports the hypothesis that this population is recovering, since morphometric and reproductive data represent young nesting turtles.

## 1. Introduction

A species life history traits are coadapted features “designed by natural selection to solve ecological problems” [1]. Studying their annual or geographic variations is crucial to understanding their ecology, how they adapt to a particular environment, and factors that determine their distributional limits and reproductive investment [2]. The life history of an organism is a series of patterns that occur throughout its life: growth, differentiation, energy storage, and principally reproduction. Organisms spend part of their life in phases of growth and differentiation prior to reproduction. Reproduction may occur once or repeatedly and may coincide with the end of the organism’s growth. However, in some cases, growth continues during the reproductive stage [3]. Fecundity and age of sexual maturity have a similar and direct influence on the number of reproductive opportunities: this combination of traits is important in defining an organism’s life history in response to a particular environment [1,4] and can change within the limits of an individual’s genotype (genetic plasticity). This plasticity determines how an organism’s genotype interacts with its environment [5,6,7,8]. When studying an organism’s life history, it is important to consider body size, growth and development rate, reproduction, clutch size, neonate size and reproductive value [1,2,4,9,10].

All sea turtle species share a specific survival strategy; they spend most of their lives in the ocean. Time spent on land is limited to when hatchlings leave the nest and make their way to the ocean and when adult females return to the beach to nest. These two life cycle stages are vital for their reproductive success and key to understanding their biology and implementing effective conservation strategies [11]. Reproduction in sea turtles (Cheloniidae and Dermochelyidae) occurs in three general forms: first, nesting can take place during conditions driven by “adult activity,” second, nesting can occur during conditions that facilitate embryonic development and survival, and third, hatchlings can emerge when conditions “benefit their survival” [12,13].

Sea turtles exhibit similar life cycles with little variation between species and populations. All species migrate from foraging areas to reproductive sites, then males return to foraging areas and females move to nesting sites. After a reproductive period of 4 to 6 months, the females return to the foraging areas and begin to prepare for the next reproductive period, which will occur within one to five years [14,15,16].

Sea turtle reproductive strategies are poorly understood due to the difficulty of conducting research in oceanic habitats where much of their life cycle occurs. Consequently, information on life history parameters such as age at sexual maturity, sexual radius, operational sex ratio, mortality, recruitment, migratory patterns, nesting intervals and philopatry is scarce [14,17,18,19,20].

Neonates have a poorly studied period of pelagic development (known as the “lost years”) [20], during which they feed and grow until they enter the juvenile stage, where they are commonly found in shallow coastal waters. The time between hatching and breeding varies for each species and population, but is probably 7 to 30 years [14,21,22].

In general, female sea turtles do not reproduce yearly, except for the Kemp’s ridley (*Lepidochelys kempii*) [14]. However, males of some species (e.g., *Caretta caretta*) can reproduce annually or biannually [23,24]. Therefore, there is no reason to assume that males and females use the same time scale for reproduction. The period between breeding seasons is known as the remigration interval. The mean remigration interval reported for female sea turtles varies by species and ranges between 1 to 9 years or more [14,22].

The remigration interval of a population may be longer than is currently reported; this discrepancy may result from tag loss, extremely long periods of inactivity, or incomplete reporting on survival. Some individuals within the population may have long periods (3–5 years) between reproductive activities [25]. Once turtles begin breeding, mature females travel to nesting beaches every two to five years to lay between two to ten clutches of eggs, at nesting intervals of nine to 15 days, depending on the species. In many populations, males appear to follow the same pattern, with mating commonly reported in waters adjacent to nesting beaches [15,26]. The two- to five-year interval between reproductive periods may be an adaptation to the high energy costs of migrating between foraging areas and nesting sites [16,27,28]. However, this pattern is also maintained in populations with shorter migrations (around 2600 km) [29,30].

Climatic conditions are the leading cause for certain species adopting a particular breeding season. Conversely, favorable conditions encourage the adults to reproduce, resulting in their offspring developing under these conditions. Reproductive cycles during the breeding season are strongly regulated internally through interactions between the hypothalamic-pituitary system and the ovaries and between various ovarian components and the neuroendocrine system of the offspring [14,31,32].

All sea turtle species have adults with large body sizes (65–180 cm), a long period of development ranging from 10 to 35 years and lay large clutches of relatively small (3–7 cm) eggs compared to other aquatic turtles [9,33,34,35,36,37,38]. Presumably, sea turtles evolved this reproductive pattern in response to high and unpredictable mortality rates during the egg and neonate life stages [9]. In addition, laying many small eggs in different clutches avoids allocating a large proportion of the female’s reproductive effort to a single progeny, which would be detrimental as individuals are likely to perish [9].

The populations of the green turtle, *Chelonia mydas*, are widely distributed throughout the Atlantic, Pacific and Indian oceans. Geographically this species is not uniform in morphological features. Carr [39] recognized two valid regional names and considered *C. mydas* as formed by two subspecies: *mydas mydas* (Atlantic, Mediterranean, Pacific and Indean oceans) and *mydas agassizii* (eastern Pacific, originally described as a full species by Bocourt [40]). This taxonomic arrangement has been supported by some authors [41], whereas other authors [42] consider *agassizii* a full species. The Chelonia populations of the east Pacific show unique morphological features, such as dark carapace pigmentation, gray plastron, narrow and high vaulted carapace and carapace indentations above the back flippers [43,44,45,46]. The adults are markedly smaller. For instance, SCL of adults in Playa Naranjo, Costa Rica is 82.9 cm, Galapagos, Ecuador 81.9 cm, and Michoacan, Mexico 77.3 cm, whereas the size of adults of populations outside the east Pacific ranges from a low of 92.2 cm in French Frigate Shoals, Hawaii to a high of 112.9 cm in Maziwi Island, Tanzania [9]. Kamezaki and Matusi [45] showed that the *agassizii* form also differ from other populations in skull morphology, and Alvarado [47] indicated differences in breeding behavior.

In spite of these differences, the results of mitochondrial DNA (mtDNA) analysis of various Chelonia populations do not support the distinctiveness of the *agassizii* form [48]. According to mtDNA results, the Chelonia populations are grouped in two clusters that correspond to the major oceanic basins: Atlantic–Mediterranean and Indo-Pacific. According to Bowen’s [48] phenogram, the samples of the eastern Pacific (Michoacan, Mexico and Galapagos, Ecuador) are included within the group of Hawaii and Oman samples.

### Black Turtle Life History Traits (Chelonia mydas agassizii)

Some of the life history traits of the black turtle (*C. m. agassizii*) have been reported for some nesting seasons of the *C. m. agassizii* population nesting at the Colola beach sanctuary, Mexico [49,50,51,52], for example, during the 1984–1985 season the average clutch size in a sample of 397 black turtle nests was 76 eggs per nest, ranging from 96 to 112 eggs. The mean number of nests laid by 379 black turtle females was 2.8 nests (range 1–9) and, the mean number of eggs laid per female per season was 213 eggs [51,52].

Based on the analysis of 907 recurrent black turtle nesters, renesting intervals of 12–14 days were reported to be the most frequent (38.5%) for the season 1984–1985 [49,50]. The mean size at which females reached sexual maturity differed between seasons within a limited range: the data suggest that black turtles reach sexual maturity at 68 cm straight carapace length. The age of sexual maturity is uncertain, and some authors suggest it occurs at 8 or 9 years. However, further research is needed with more reliable data [23,24,52]. For animals with long life cycles, such as sea turtles, delaying the start of reproduction to concentrate on growth and reaching a large size may be an effective strategy to ensure long-term reproduction. For a female, egg production and long migratory routes to and from nesting sites significantly reduce their energy reserves; therefore, growth must be slowed or suspended. If obtaining a large size translates into more efficient protection against natural predation, then female *C. m. mydas* may sacrifice early reproduction to reach a larger size in less time [51].

In this study, we provide updated information on the life history traits (body size, clutch size, egg size, fecundity, remigration interval, sexual maturity and growth rate) of the largest eastern Pacific black turtle population nesting in Colola Beach, Michoacan, Mexico. Updating information on life history traits makes it possible to identify changes in reproductive and morphological traits over time that characterize a given population that may be caused by variations in selection pressures that affect it (changes in habitat, food availability, environmental stochasticity, etc.) to support management decisions and conservation measures of this morphologically and reproductively unique *C. m. agassizii* population.

## 2. Materials and Methods

To determine black turtle life history traits for the nesting population in Colola, Mexico (18°30′0′-18°0′0′N, 103°40′0′-102°50′0″W) (Figure 1), morphometric and reproductive annual data from 15 nesting seasons (1985–2000) which included between 3 and 5 remigratory events (the 1992 season was not included due to insufficient data) were obtained from the Black Turtle Ecological Recovery Projects database for Colola Beach, Mexico.

Data collection during the 15 study seasons began in September of each year and ended in January of the following year. Nesting females were tagged with plastic and Monel steel tags [53]. The mean annual values were obtained for carapace size (CCL), clutch size, egg size, fecundity, remigration interval, age of sexual maturity and growth rate. These life history traits were selected as they are the most commonly studied for sea turtles.

### 2.1. Statistical Analyses

Body size was recorded by measuring with a flexible tape measure (scale 0–150 cm) the curved carapace length (CCL) of 100 nesting females per season (n = 1500), with a caliper (scale 0–127 centimeters), straight carapace length (SCL) from a sample of 100 females, and an analysis of variance (ANOVA) and Tukey’s test of differentiation of means were used to analyze the interannual variations in female body size. Samples of 100 females per season were randomly selected considering nesting females throughout the nesting season (September–January). In both, the CCL and the SCL, the standardized measurements of standard carapace length, were recorded.

Clutch size was determined through the analysis of 1,500 clutches corresponding to 15 black turtle nesting seasons, with a random sample of 100 clutches of the same number of females taken for each season (September–January). The interannual clutch size variation for the same period was also determined through analysis of variance (ANOVA). Maximum egg diameter and egg weight (wet mass) after oviposition were examined with a mechanical caliper for a random sample of 20 eggs per nest from 100 nests (28.8% of total of clutch size). Pearson’s correlation coefficient and linear regression analysis were applied to determine the relationship between the diameter and weight of the egg and the female size (CCL).

Mean fecundity was estimated based on a sample of 700 females that nested from 1986 to 2000 (n = 50 per season). Significant differences between seasons were analyzed using a Kruskal–Wallis analysis of variance. In addition, Tukey’s test of differentiation of means was used to identify differences between years.

### 2.2. Mathematical Calculations

Female remigration intervals were determined for a total of 460 females from 1985 to 2000, for which renesting information was available in years after they were first tagged on Colola Beach, Mexico. All annual records (1985–2000) of remigrant turtles tagged from 1985 and recaptured in later years were analyzed.

To determine age of sexual maturity, the following equation was used: SM = ((SCL − 40 cm)/1.4 cm)) + 10 years, where SM is the age of sexual maturity reached at a given size considering a constant growth rate of 1.4 cm/year from the age of recruitment of juveniles to coastal development habitats. SCL is the straight length of the carapace in centimeters. Forty corresponds to the centimeters of SCL that a turtle reaches at the age of 10 years when juveniles are recruited from pelagic habitats to coastal habitats. The 1.4 cm is equivalent to the rate of growth presented by the turtles per year in the feeding areas. Finally, 10 years are added, which is equivalent to the estimated recruitment age from the pelagic stage to coastal habitats at a size of 40 cm SCL.

The mean growth value (1.4 cm × year^−1^) reported for this species in Baja California [44] was used to determine the sexual maturity of the *C. m. agassizii* population that nests in Colola, Mexico, and the mean growth data reported for other populations of *C. m. mydas* in the world. Growth data were extrapolated to the nesting population of Colola, Mexico, from which carapace length values were previously obtained (n = 1500), referring to females nesting for the first time. Also, the minimum size and mean SCL (n = 194) were estimated to determine the minimum size at which females laid their first clutch. The mean yearly growth rate was determined (n = 25) using the mark–recapture method for remigrant females of 3–11 years.

The mean growth of reproductive females was estimated based on information from 25 remigrant females measured in seasons prior to the recapture date.

## 3. Results

The average number of nesting females per season for the study period was 430 and the annual sample of 100 females corresponded to 23.2% of the nesting females.

### 3.1. Body Size

The mean size of the nesting females was 85.71 cm CCL (range = 60–110 cm; SD ± 6.831; n = 1500) (Figure 2). The mean of SCL was 77.9 cm (range = 64.2–93.4 cm; SD ± 5.31; n = 100 females).

The analysis of variance showed significant differences in the CCL (F = _14 gl_,1485 = 15.04, *p* = < 0.05). Tukey’s mean difference analysis showed significant differences between turtle sizes in 1999 and all other years (*p* = < 0.05) (Figure 3).

### 3.2. Clutch Size

A mean clutch size of 69.3 eggs per clutch (n = 1500; range: 3–150; SD ± 23.01) was found (Table 1).

The analysis of variance (ANOVA) showed that during the study period (1985-2000), there were significant differences in clutch size between seasons (F = _14,1485_ = 7.685, *p* < 0.0001).

Tukey’s mean differentiation analysis showed significant differences between the mean clutch size of 1999 and the rest of the seasons (*p* < 0.001) except for 1985 and 1988, for which there were no significant differences (*p* > 0.001). The 1988 season also presented significant differences (*p* < 0.001) from 1989 and 1987 (Figure 4).

### 3.3. Egg Size

The mean egg diameter found for the population of *C. m. agassizii* was 42.1 mm (range = 36.0–49.0 mm; SD ± 19.0 mm) (n = 2000 eggs, 28.8% of total of eggs). No correlation was found between egg diameter and female curved carapace length (*p* > 0.05). The mean egg weight was 41.5 g (range = 30.5–53.7 g; SD ± 4.62 g). No significant relationship was found between mean egg weight and female curved carapace length (*p* > 0.05). A linear relationship was found between clutch size and curved carapace length (R^2^ = 0.4231; r2 = 0.1790; *p* < 0.0001). Although the correlation is positive, the relationship between carapace size and clutch size should be taken with caution due to the low value of R^2^ (Figure 5).

### 3.4. Fecundity

Mean fecundity was 196.4 ± 91.2 eggs per female per season (range 38–784 eggs per female per season). The Kruskal–Wallis test showed significant differences between the interannual average fertility of 1986–2000 (H-test = 61.1, df = 13, *p* = 0.0001). Tukey’s mean differentiation test showed significant differences between the mean fecundity of the 1999 season and the fecundity of the following nesting seasons: 2000 (*p* = < 0.05); 1997 (*p* = 0.001); 1995 (*p* = < 0.05); 1993 (*p* = 0.001); 1988 (*p* = < 0.05); 1986 (*p* = < 0.05). The 1994 season presented significant differences from the 2000 season (*p* = < 0.05) (Figure 6).

### 3.5. Remigration Interval

A total of 528 nesting female remigration events were registered between 1985 to 2000 with intervals of 1 to 11 years. The most frequent interval was three years (23.2%), followed by intervals of four years (18.9%), five years (17.4%) and six years (11.7%) (Figure 7), with the rest having remigration intervals of one, two, seven, eight, nine and ten years (21.2%).

Female carapace size (CCL) presented significant differences between individuals with reproductive cycles of two to six years (F = 3.213 DF:4288, *p* = 0.013). Females with remigration intervals of two years presented a mean size of 89.2 cm (n = 37), whereas females with remigration intervals of three and five years presented a mean size of 85.6 cm (n = 74) and 85.0 cm (n = 66), respectively. Regarding fecundity, there were no significant differences between females with remigration intervals of two to six years (statistical test = 3.589 DF = 4.289 *p* = 0.464). Similarly, the mean clutch size did not present significant differences between reproductive cycles (F = 1,366 DF = 4292, *p* = 0.246). Table 2 presents the remigration events for each year (1985–2000).

### 3.6. Sexual Maturity

The mean curved carapace length (CCL) of the black turtle in Colola, Mexico, was 85.7 cm, with a minimum size of 60 cm (n = 1500). Straight carapace length (SCL) was 80.0 cm. Using the mean growth rate (1.4 cm × yr^−1^) and minimum size (SCL 40 cm) reported in Baja California for juveniles [45] and considering that at least ten years is required to reach this size [46], then the minimum age at which female black turtles reach sexual maturity is 24.2 years if the growth rate remains constant. Using the same growth rate (1.4 cm × yr^−1^) and the mean SCL of 80.0 cm, female turtles at Colola Beach have a mean age of 38.6 years.

Based on the differences in size found in 25 remigrant females, mean growth was 0.55 ± 0.67 cm year^−1^ (range 0–1.9 cm × year^−1^ n = 25).

## 4. Discussion

According to the results obtained in this work, the black turtle population presents the lowest life history traits compared to those reported in the literature in populations outside the eastern Pacific region. Female black turtles (*C. m. agassizii*) from the eastern Pacific nesting population are the smallest when compared to other green turtle populations worldwide [16]. As with other populations of *C. m. mydas*, the eastern Pacific black turtle presents a positive relationship between body size and clutch size; however, this correlation should be taken with caution due to the low value of r^2^. This coincides with Buskirk and Crowder’s [9] findings for all sea turtle species. The majority of nesting females (n = 60%) presented a CCL ranging from 80 to 90 cm, which is relatively large, yet smaller than those for other green turtle populations reported by Hirt [16]. As many life history traits are related to body size, the reduction in the mean size of black turtles in the nesting population due to the consumption of large adults in northern Mexico could have affected fecundity, clutch size, and hatchling size. It has been reported that larger turtles generally produce more nests per season [21]. On the other hand, Buskirk and Crowder [9] report that in the seven species of sea turtles, many of the reproductive characteristics vary with body size: large species tend to lay large eggs that produce large hatchlings. Large species also lay more clutches per year and invest significant reproductive effort. In addition, the demographic bottleneck caused by intense exploitation in the 1960s and 1970s could have, in turn, created a genetic bottleneck in the population [54,55,56], which may be determining the life history traits of the breeding population in the eastern Pacific.

The mean clutch size for turtles nesting at Colola is lower than that of the 26 *C. m. mydas* populations reported by Buskirk and Crowder [9]. Clutch size is one of the most important life history traits, affecting the species’ survival and hatchling fitness. The clutch size for *C. m. agassizii* is the smallest reported for *C. m. mydas* populations globally. The eggs’ mean maximum diameter was lower than that reported by Hirt [16] for 18 populations of *C. m. mydas*, which presented a maximum diameter of 58.7 mm, almost 10 mm greater than that found for the black turtle population in Colola, Mexico (mean: 45 mm; range = 33.8–58.7 mm). The mean egg weight reported for 16 populations of *C. m. mydas* is 47 g. (range = 21–66 g).

It was observed that the body size of the females influenced fecundity. In the 1999 season, the average size of the females was 92.4 cm CCL and this was reflected in the increase in the fecundity of the females.

We found that larger females exhibit shorter remigratory periods (two years) and those of smaller size exhibit longer remigratory periods (three and five years). It is likely that larger turtles have a greater ability to obtain energy resources for reproduction that allows them to perform reproductive migrations in short periods. Among the morphological adaptations of herbivorous reptiles is body size. Iverson [57] has suggested a number of hypotheses to explain this phenomenon: 1. a large body has greater mechanical strength for the utilization of food of fibrous consistency, 2. this reduces competition and predation, 3. as the metabolic rate is lower the larger the reptile is, it allows them to survive using a difficult-to-digest and energy-poor food source (vegetable products), 4. large animals can use more of an ephemeral food source that is only available for short periods, and 5. large animals can store more food during periods of abundance and can probably withstand longer periods of food shortage. According to the results of this work, in relation to the variations in body size and the remigration periods of three and five years, it can be hypothesized that female black turtles have differential access (qualitatively and quantitatively) to food resources in feeding areas in northwestern Mexico (Baja California) and Central America. It is likely that females feeding in Central America have more restrictions in food abundance due to environmental variations caused by the El Niño current (ENSO), which would cause lower growth rates in juveniles and longer remigratory periods (five years) in relation to females feeding in Baja California; however, detailed studies are needed to verify this hypothesis.

The population of black turtles that nests in Michoacán reaches sexual maturity at a minimum size of 60 cm SCL and age of approximately 24 years. However, the mean age at which females nest is 38.6 years, considering a mean growth rate of 1.4 cm/year, as reported for the population found on foraging grounds in Baja California [10].

## 5. Conclusions

The study of the life history of the black sea turtle and its interannual variations contributes to the understanding of the biology of *C. m. agassizii*, which allows definition of the mechanisms (differences in reproductive investment in sea turtles depending on numerous endogenous factors (genetics, age, body size, health status, and reproductive history) and exogenous factors (migratory distances, latitude of feeding grounds, and quality of foraging areas) that have shaped its life strategy and allows us to prevent a decline in populations and design conservation strategies according to the increasingly rapid and unpredictable changes in the development and reproductive habitats of this population. According to the values of life history traits found in the breeding population of Michoacan, the morphometric and reproductive traits are the smallest compared to those reported in the literature for different populations of *C. m. mydas* in their pantropical distribution. The particularities of the life history traits found in the black turtle population seem to be a consequence of adaptive processes to the particular environmental variations of the region, such as those caused by the El Niño phenomenon (ENSO), which have shaped the life history strategies of the population, Therefore, we consider it important to continue monitoring these traits in order to identify variations in morphological and reproductive aspects (e.g., body size, clutch size and fecundity) that could affect the recovery and survival of the population in the long term. Therefore, the black turtle population should be considered a conservation priority in the eastern Pacific region.

## Figures and Tables

**Figure 1 animals-13-00406-f001:**
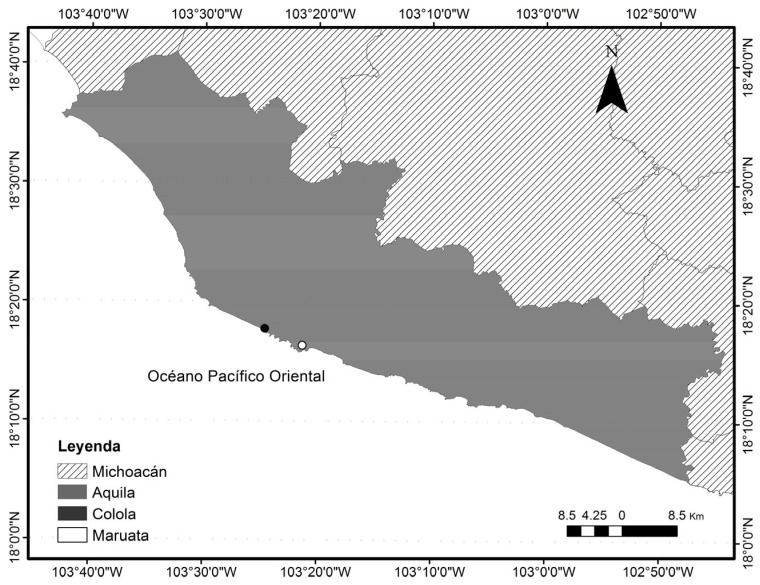
*C. m. agassizii* nesting area in Colola, Mexico.

**Figure 2 animals-13-00406-f002:**
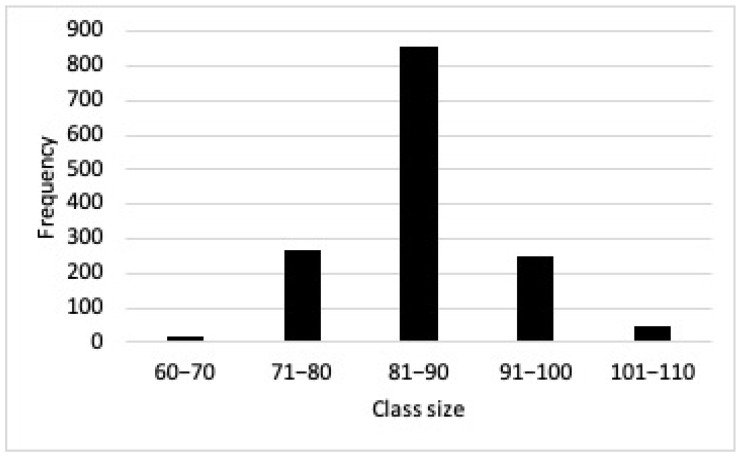
Curved carapace length (CCL) of 1500 nesting females (1985–2000).

**Figure 3 animals-13-00406-f003:**
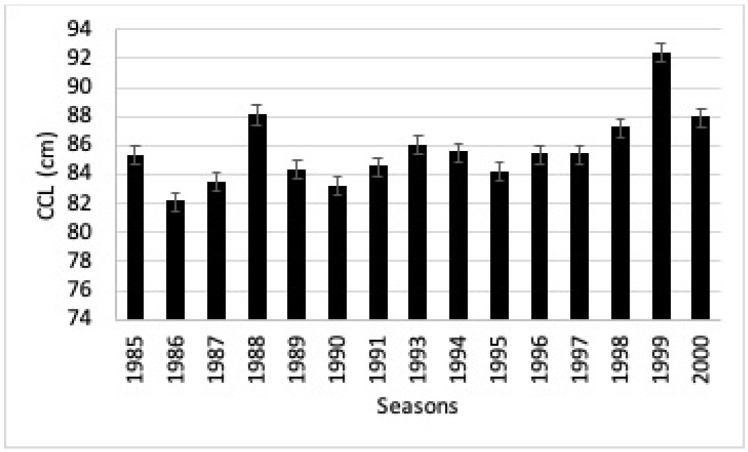
Temporal changes in body size (CCL) of black turtles (1985–2000).

**Figure 4 animals-13-00406-f004:**
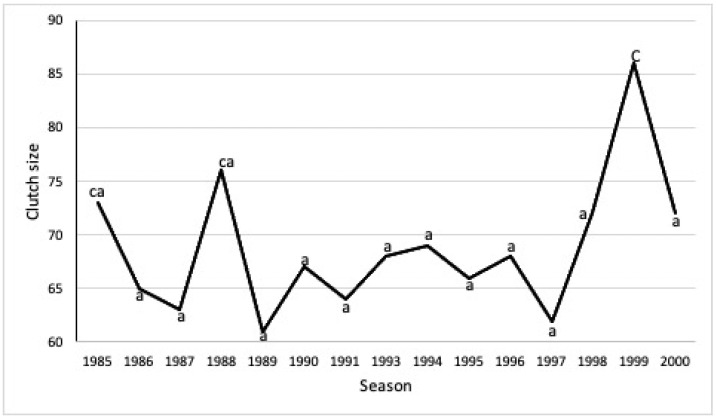
Mean clutch size per year 1985 to 2000 (n = 1500). The letters indicate significant differences between the years analyzed.

**Figure 5 animals-13-00406-f005:**
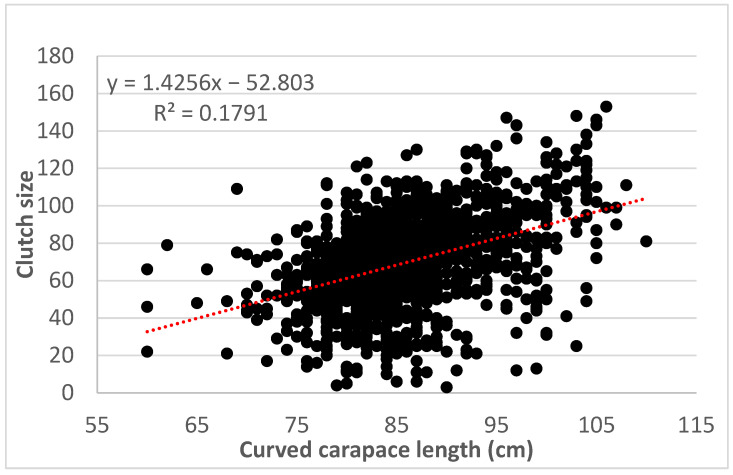
Linear regression between female curved carapace length and clutch size for *C. m. agassizii* (n = 1500) at Colola Beach, Mexico.

**Figure 6 animals-13-00406-f006:**
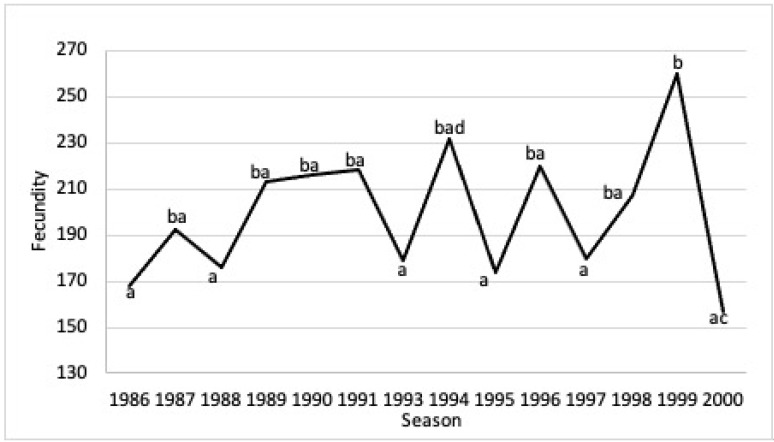
*C. m. agassizii* mean fecundity during 14 nesting seasons (1986–2000) at Colola Beach, Mexico. The letters indicate significant differences between the years analyzed.

**Figure 7 animals-13-00406-f007:**
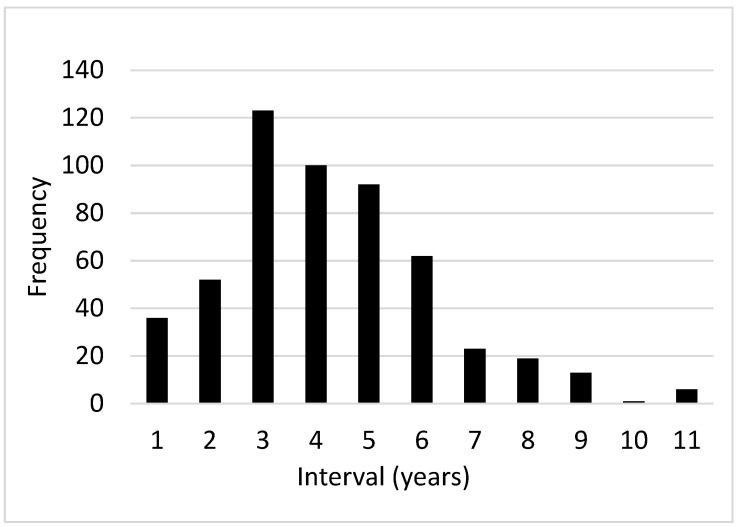
*C. m. agassizii* female remigration interval (1985–2000).

**Table 1 animals-13-00406-t001:** Statistical data for clutch size from 1985 to 2000 (n=100 nests per season).

Season	2000	1999	1998	1997	1996	1995	1994	1993	1991	1990	1989	1988	1987	1986	1985
**N**	100	100	100	100	100	100	100	100	100	100	100	100	100	100	100
**Minimum**	10.0	12.0	20.0	4.000	17.0	17.0	11.0	20.0	16.0	25.0	3.0	21.0	5.0	11.0	18.0
**Maximum**	130.0	148.0	146.0	118.0	113.0	123.0	153.0	118.0	124.0	120.0	130.0	136.0	111.0	127.0	118.0
**Average**	*72.6*	*85.6*	*72.6*	*63.8*	*68.4*	*66.6*	*69.6*	*68.6*	*64.6*	*67.6*	*61.7*	*76.5*	*62.8*	*65.5*	*73.6*
**Standard deviation**	25.2	26.9	25.5	18.3	21.7	17.8	25.3	19.7	18.4	17.9	24.5	24.6	19.4	23.8	21.9

**Table 2 animals-13-00406-t002:** Remigration events of 528 female *C. m. agassizii* tagged in Colola, Mexico and recaptured in subsequent years (n = females recaptured per season).

	1985	1986	1987	1988	1989	1990	1991	1992	1993	1994	1995	1996	1997	1998	1999	2000
**1985**		5	5	8	10	7	1									
**1986**			11	5	14	17	2				3					
**1987**				5	10	30	24	21	9	8	9	3		5		
**1988**					1	18	4	5	2	3	3	3	1			
**1989**						3	5	9	10	23	36	6	7	4		
**1990**							2	1	6	11	27	10	3			
**1991**								1	2	10	11	6	3	3		
**1992**										1	1	3	1	1		
**1993**										2	6	11	4			
**1994**											1	4	14	3		
**1995**												3	4	2	2	
**1996**													1	5		1
**1997**														1	1	
**1998**																3
**1999**																
**2000**																

## Data Availability

The data presented in this study are available on request from the corresponding author.

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
