# Peer review of "Black Sea Turtle (Chelonia mydas agassizii) Life History in the Sanctuary of Colola Beach, Michoacan, Mexico"

_animals, 2023, doi:10.3390/ani13030406_

Round 1

Reviewer 1 Report

Dear Author(s),

The manuscript you have presented on the 15-year life history of the Eastern Pacific Green Turtle (EPGT) on the beach of Colola Beach, Michoacan, Mexico, provides valuable data.

However, I don't think the research has a hypothesis. It goes no further than contributing to the current life story of EPGT. It is not clear what kind of new information will contribute to the current situation. Because, as you mentioned in the "Introduction" section, some information about the life history of EPGT (egg size, nesting interval, conception frequency, fertility) has already been given by the authors. You have very valuable data that goes back many years. However, the information you present is already available in the literature, what do you present as new information? I suggest you look at it from a different perspective and develop a new hypothesis (eg, trend analysis of long-term (15-year) reproductive output and morphology).

Introduction, MM and result section of the presented manuscript could have not been synchronized. For example, while one of the purposes was to evaluate the number of nests, the results on this subject were not included at all. In addition, morphological analysis is not given in the purposes, but is explained in the results section. I strongly suggest that the manuscript be rewritten more carefully and in sync between chapters.

You can see my detailed suggestions about the manuscript on the PDF copy. However, in summary, it is as follows:

Introduction;

- It is important that you provide brief information about the taxonomic status of EPGT.

-I suggest that you reconsider the manuscript's hypothesis and reassess the data over many years (eg, trend).

MM

-Add a tag protocol.

-how did you determine the age to reach sexual maturity?

-The CCL measurement the minimum CCLmin or CCLmax?

-Why you only chose 100 nests. What is the ratio of these 100 nests to the total nests, ie what percentage?

-You can give statistical analyzes in a separate sub-title.

Results

 Mention just your own result in "Result" section.

Write the species names in italics and short.

There is no data on the number of slots.

Discussion

What do you think might be causing it to be the smallest? (feeding ground, long migration route, high mortality rate, genetics?)

I'm sorry, I hope the suggestions I've made will help you do a higher-impact study. However, I am optimistic about the manuscript, as I think that your valuable data will contribute to the population dynamics of EPGT and to reveal its conservation and management plans. Therefore, I recommend it as a "Major revision" for reconsideration.

Best Wishes

Author Response

Dear reviewer,
Please see the attached file. 
Kind regards 
Cutzi Bedolla-Ochoa

Reviewer 2 Report

This paper contributes to the understanding of life history traits of the black turtle Chelonia mydas agassizzi at one of the most important sites in the eastern Pacific. It provides evidence that these traits differ from other Chelonia populations in the world and highlights the importance of long-term monitoring for their evaluation.

However, the authors felt short in discussing their findings, their cause and how these affect the Chelonia mydas agassizzii population. They did a great job describing the life history traits and how these compare to other Chelonia populations but not the implications for their recovery and conservation. 

In the last 5 years, several published papers have discussed the changes in life history traits in several species and how these affect their recovery. I strongly suggest the authors to review this literature to strengthen their discussion and conclusions.

Specific comments

Lines 26-29: This statement is a bit flawed. Studies have shown that east Pacific green turtles (black sea turtles) are smaller and lay fewer eggs than green turtles in other regions. Therefore, these differences do not support your hypothesis of a recovering population. Furthermore, almost all green sea turtle populations are in a recovering phase and show size reductions.

Lines 59-73: The information in these paragraphs seems redundant. Authors should be able to write this more succinctly to avoid repetition and facilitate the reading.

Line 75: The scientific names should be written in italics to respect the nomenclature.

Lines 106-112: It is a bit unclear if the data presented was obtained during the 1984-1985 nesting season or this was just for the re-nesting intervals. On the other hand, it would be useful to know when this data was collected, particularly if the paper is trying to update the current knowledge on the black sea turtle.

Lines 125-128: What was the purpose of the analyses? This is a key element of the manuscript and is missing.

Lines 144-146: Is 100 nesting females per nesting season a representative sample? What were the criteria of selecting this number? The same question applies to the number of clutches selected.

Lines 146 and 158: One disadvantage of doing multiple comparisons is that it could give significant results even when there is none. I suggest conducting a Bonferroni correction to reduce the risk of false significance.

Lines 156-158. Are you saying that the variation of mean fecundity between seasons was analyzed using a Kruskal-Wallis test? If so, please change the wording.

Line 188. The scientific name Chelonia mydas should be in italics.

Lines 195-197. Could the authors please explain what were the significant differences? Was the mean clutch size higher or smaller in the years that were significantly different?

Line 203. The scientific name should be in italics.

Line 208. Was the linear relationship positive or negative? On the other hand, I would be cautious with this statement. Although the p-value is significant, the values of r and r2 are too low to suggest a relationship.

Line 216-218: What was the nature of these differences? In other words, did the mean fecundity increase or decrease between nesting seasons?

Line 256: The scientific name has to be in italics.

Line 257-258. This is not true, the r and r2 values are too low to say there is a positive linear regression, even if the p-value is significant.

Lines 260-263. This is a bit confusing. Are you saying that in 1999 most of the females (60%) had CCL ranging from 80 to 90 cm? or is this percentage for the 15 nesting seasons? It is not clear what was the difference between 1999 and the other 14 nesting seasons.

Lines 264-266. Please, explain how the consumption of large adults affected fecundity, clutch size and hatchling size and provide a citation.

Line 268. How is the genetic bottleneck determining the life history traits of this population?

Lines 270, 273, 275, and 277: The scientific names should be written in italics.

Lines 271-274: If this is completely true, then why is the population recovering so well?

Lines 282-284. This sentence contradicts the one before where you say that 38.6 is the mean age. Furthermore, this value is considering all the females measured during the 15 years of study, correct? But you evaluated size distribution per year, does the mean of 38.6 years is the most observed value between seasons?

Line 288-290. But what are those mechanisms? You have not explain how these traits affect the recovery of the population.

Last comment, based on the data you are providing in figures 3, and 5, perhaps using other type of graph such as confidence intervals or box and whiskers would be better to present your results.

Author Response

Dear reviewer,
Please see the attached file. 

About

Line 268- During the 60's and 70's the capture of black turtles was directed mainly to larger individuals because they were sold by weight and not by individual, this probably resulted in a reduction in the size of the breeding adults, with smaller individuals progressing.

Lines 271-274: Possibly this was due to the fact that as size decreases, the metabolic rate per unit weight increases, which may result in a decrease in generation time and consequently an increase in the growth rate of the population. In sea turtles many of the reproductive characteristics vary with body size which influences newborn size and hatchling size. 

Kind regards 
Cutzi Bedolla-Ochoa

Round 2

Reviewer 1 Report

Dear author(s);

There are very few differences between the previous version of MS and the current version. Many points had not been clarified by you. I made the explanations on the pdf version of MS. I also list the important points below;

Introduction

- Line 104: It would be nice if you provide a brief description of the taxonomic status of Chelonia mydas agassizii within the framework of taxonomic discussions.

- line 106: What do you mean by Chelonia? Thinking black and green together? Or just black? Could you please explain? Also use the same rhetoric throughout MS. Because it gets confusing.

- Line 105: Please use the names "Black turtle" for "Chelonia mydas agassizii" and "Green turtle" for "Chelonia mydas".

- Line 127-131: I strongly recommend that you take a different perspective on the purpose of MS and develop a new hypothesis (eg, long-term (15-year) reproductive output and trend analysis of morphology).

However; if you want to maintain the hypothesis based on more than 15 years of life history data, you can build MS on the need to "update" the life history data.

MM

- I recommend using a separate paragraph for statistical analysis and mathematical calculations. Thus, readers can clearly see which test is used for what purpose and how mathematical calculations are performed.

- Line 150: Please cite to the protocol (CCLmin or CCLmax) you used for morphological measurements.

- Lines 141-142: Calculate separately under each sub-title (eg body size, egg size, clucth size).

-Lines 171-181: Specify the mathematical calculation method for sexual maturity.

Results

Line 185-187: No statistical test results? How has it changed over the years? Or in what years did the differences arise? It would be nice to see the average CCL or SCL change over the years in the form of a column chart or a table.

Also, the number of females for which the results are given corresponds to what percentage of the total females.

-Line 191: The number of clucth sizes in which the results are given corresponds to what percentage of the total clucth size.

Line 208. Please provide descriptive statistics for egg size data. How many eggs were measured in total? The measured egg corresponds to what percentage of the total egg?

Line 223: Please provide descriptive statistics of fecundity data.

Line 228: Cluct size or fecundity? Please clarify

Line 272: Does 24.2 years is your own result? How did you calculate?

Line 273: Does 38.6 years is your own result? How did you calculate?

Conclusions

Line  335-338: What advantages or disadvantages might this situation have for the future? What are the threat(s) that will affect the population? What precautions should be taken? What new topics should be researched?

Author Response

Dear reviewer, attached file for review
Kind regards

Cutzi Bedolla

Reviewer 2 Report

Dear authors, thank you for reviewing your manuscript and considering some of the suggestions made. I did not find a response document to my suggestions and I would appreciate if you can provide one to understand your decision of accepting or declining them.

I still have some concerns regarding the materials and methods, the discussion and conclusion sections, particularly in the following:

Line 114: When were the 907 nesters recorded? Was this in the 1984-1985 period? Please, specify.

Lines 127-131: The purpose of your research is key and how you describe it can make your paper very interesting or not. At the moment, it does not reflect the value of your research because it is not saying anything new about the black turtle. I suggest rewording this paragraph to something like this: "In this study, we provide updated information on the life history traits (clutch size, fecundity, reproductive cycles, age at sexual maturity, egg size and the number of nests laid per season) of the largest Eastern Pacific black turtle population nesting in Colola Beach, Michoacan, Mexico, to support management decisions and conservation measures of this morphological and reproductively unique Chelonia population."

Lines 132-135. This sentence should be in the materials and methods section.

Lines 151-154:  As I mentioned before, multiple comparison tests can give significant results even when there is none. I suggested conducting a Bonferroni correction to reduce the risk of false significance. What was your reasoning for not doing one? Were the tests corrected in any other way? The same goes for clutch size and mean fecundity.

Line 164-166: This sentence needs rewording: Mean fecundity was estimated based on a sample of 700 females that nested from 1986-2000 (n=50 females per season). Significant differences between seasons were analyzed using a Kruskal-Wallis analysis of variance.

Line 280: It should say Eastern Pacific.

Lines 286-287: there is more recent literature on size at maturity for Chelonia mydas than the one cited. I encourage the authors to review and revisit your discussion. 

Lines 288-294. Thank you for your response to my comment. I suggest to include your explanation in the manuscript so other readers can understand your point.

Lines 295-297. Interesting point but you have not explained how the bottleneck is determining the life history traits. I assume it has to do with the selection of smaller individuals (those that were not hunted) and their genetic composition. But, what is the effect? are the life history traits decreasing with time?

Line 315: The correct citation is Iverson.

Lines 329 -336. Thank you for considering my comments. However, the added text makes your conclusions harder to read and understand. I suggest reorganizing your ideas for this section according to the main points of your work: 1) life history traits of black turtles in Colola are the smallest compared to other Chelonia populations. 2) How this study contributes to the understanding of the biology of Chelonia. 3) What are the conservation implications of your findings.

Line 369. This citation needs to be updated. It says "en prensa" but it has been published already.

Table 1 has some typos, please change "Maxumum" to Maximum and "desviation" to deviation.

Overall comments: throughout the text and the literature cited, the scientific names and the genus should be in italics.

Author Response

Dear reviewer, attached file for review

Regarding your kind suggestion to use the Bonferroni analysis, I would like to mention that we analyzed the option and concluded that due to the large number of means involved, we consider that the Tukey test has more statistical strength than Bonferroni.

Kind regards

Cutzi Bedolla

Kind regards

Round 3

Reviewer 1 Report

Dear Authors,

First of all, thank you very much for considering my suggestions.  I suggested minor changes in MS word format. It would be great if you could write a few sentences about the importance of updating of life history data, especially in the introduction, to support the MS hypothesis. 

Also, I am very apolagize to late write, I may have missed it in the previous version. I made a few suggestions for morphological adaptations in the discussion section.

Please note the abbreviations throughout the MS. Also, add a few sentece about data homogeinety.

best wishes

Author Response

Dear reviewer,
I attach to this email the file with the suggested corrections which are marked with red color.
Best regards
Cutzi Bedolla and authors
